# Oxidative Stress and Psychiatric Disorders: Evidence from the Bidirectional Mendelian Randomization Study

**DOI:** 10.3390/antiox11071386

**Published:** 2022-07-18

**Authors:** Zhe Lu, Chengcheng Pu, Yuyanan Zhang, Yaoyao Sun, Yundan Liao, Zhewei Kang, Xiaoyang Feng, Weihua Yue

**Affiliations:** 1Peking University Sixth Hospital, Peking University Institute of Mental Health, Beijing 100191, China; luzhe@bjmu.edu.cn (Z.L.); puchengcheng@bjmu.edu.cn (C.P.); zhang_yyn@bjmu.edu.cn (Y.Z.); sunyaoyao@bjmu.edu.cn (Y.S.); yd.liao@bjmu.edu.cn (Y.L.); kangzhw@bjmu.edu.cn (Z.K.); xiaoyangfeng@bjmu.edu.cn (X.F.); 2NHC Key Laboratory of Mental Health, Peking University, Beijing 100191, China; 3National Clinical Research Center for Mental Disorders, Peking University Sixth Hospital, Beijing 100191, China; 4PKU-IDG/McGovern Institute for Brain Research, Peking University, Beijing 100871, China; 5Chinese Institute for Brain Research, Beijing 102206, China

**Keywords:** oxidative stress, psychosis, Mendelian randomization study

## Abstract

Observational studies have shown that oxidative stress is highly related to psychiatric disorders, while its cause–effect remains unclear. To this end, a Mendelian randomization study was performed to investigate the causal relationship between oxidative stress and psychiatric disorders. On the one hand, all causal effects of oxidative stress injury biomarkers (OSIB) on psychiatric disorders were not significant (*p* > 0.0006), while the findings suggested that part of OSIB was nominally associated with the risk of psychiatric disorders (causal OR of uric acid (UA), 0.999 for bipolar disorder (BD), and 1.002 for attention-deficit/hyperactivity disorder (ADHD); OR of catalase was 0.903 for anorexia nervosa (AN); OR of albumin was 1.162 for autism; *p* < 0.05). On the other hand, major depressive disorder (MDD) was significantly associated with decreased bilirubin (*p* = 2.67 × 10^−4^); ADHD was significantly associated with decreased ascorbate (*p* = 4.37 × 10^−5^). Furthermore, there were also some suggestively causal effects of psychiatric disorders on OSIB (BD on decreased UA and increased retinol; MDD on increased UA and decreased ascorbate; schizophrenia on decreased UA, increased retinol and albumin; ADHD on increased UA, and decreased catalase, albumin, and bilirubin; AN on decreased UA). This work presented evidence of potential causal relationships between oxidative stress and psychiatric disorders.

## 1. Introduction

Psychiatric disorders, with great heterogeneity in symptoms, pathogenesis, prognosis, and resource allocation, are some primary public health issues across the world. The Global Burden of Disease, Injuries, and Risk Study in 2016 confirmed that mental and substance use disorders, led by major depressive disorders (MDD), were major causes of non-fatal burdens, which have profound impacts on individuals and society [1]. Despite the numerous studies on psychiatric disorders, the etiology and pathogenesis remain unknown. Generally, the genesis of psychiatric disorders is a result of dysfunction of dopaminergic, serotoninergic, and glutamatergic neurotransmission. However, some other components are involved, such as neurotrophic factors, the immune system, neuroendocrine system, and epigenetics. In recent years, an increasing number of studies have emphasized the role of oxidative stress in the pathophysiology of psychiatric disorders [2].

It is known that a balanced oxidative stress status is essential for the normal functioning of the body. A high level of oxidation might lead to oxidative changes in proteins, which play important roles in many human diseases [3]. The excessive level of peroxides leads to deleterious oxidation and chemical modification of biomacromolecules, consequently contributing to various pathological mechanisms of diseases. The brain is a lipid-rich organ, with enormous oxygen consumption, and a lack of sufficient antioxidant barriers, which makes the brain highly susceptible to oxidative stress imbalance. Therefore, unbalanced oxidative stress is linked to the pathophysiology of various psychiatric disorders, including schizophrenia (SCZ), MDD, bipolar disorder (BD), and so on. More, redox homeostasis is ensured by a complex antioxidant defense system, including enzymatic and non-enzymatic antioxidants [4]. Enzymatic antioxidants are the main antioxidant systems in the cell, mainly covering glutathione transferase (GST), catalase (CAT), superoxide dismutase (SOD), and glutathione peroxidase (GPX). Non-enzymatic antioxidants include the principal antioxidant system in extracellular fluid, such as glutathione (GSH), vitamins (e.g., vitamin A, vitamin E, and vitamin C), uric acid (UA), albumin, total bilirubin (TBIL), and some metal ions, such as zinc [5,6].

It was noted that there are numerous abnormalities of antioxidant defense in patients with psychiatric disorders. Compared with healthy individuals, patients with SCZ have lower GSH, total antioxidant status, docosahexaenoic acid levels, SOD, GPX activity, higher homocysteine, interleukin−6, thiobarbituric reactive substances (TBARS), nitric oxide (NO), and tumor necrosis factor-alpha [7,8,9]; patients with BD are associated with higher GST, higher CAT, higher nitrites, higher TBARS, malondialdehyde, UA, and lower GSH [10,11]; patients with MDD are associated with higher MDA, higher SOD, and lower UA serum, paraoxonase, albumin, and zinc levels [12,13]; obsessive-compulsive disorder (OCD) patients have higher 8-hydroxydeoxyguanosine, MDA, GPX, SOD, and lower total antioxidant status, vitamin C, vitamin E [14]; anorexia nervosa (AN) patients have higher CAT and NO [15]. In addition, previous studies showed that the oxidative status changed after treatment. For instance, antidepressant treatment can significantly reduce MDA and increase UA and zinc levels [12,13]; oral re-alimentation could lead to increased albumin and apolipoprotein B reduction [16]. Antioxidative treatment may alleviate some symptoms of psychiatric disorders, such as SCZ [17,18], MDD [19], and ASD [20].

Moreover, it remains controversial as to whether such an oxidative stress injury is a cause or a downstream effect of psychiatric disorders. In such cases, the Mendelian randomization (MR) study provides the genetic variants from genome-wide association studies as instrumental variables (IVs) for environmental exposure to make causal inferences about the outcome [21]. To unveil the causal associations between oxidative stress injuries and psychiatric disorders, a two-sample bidirectional MR study was conducted.

## 2. Materials and Methods

### 2.1. Study Design

A bidirectional MR design was applied to detect the causal effects of 11 oxidative stress injury biomarkers on 7 psychiatric disorders (Figure 1). A group of oxidative stress injury biomarkers was composed of superoxide dismutase (SOD), glutathione S-transferase (GST), glutathione peroxidase (GPX), catalase (CAT), uric acid (UA), zinc, alpha-tocopherol, ascorbate, retinol, albumin, and TBIL); psychiatric disorders included autism spectrum disorder (ASD), attention-deficit/hyperactivity disorder (ADHD), schizophrenia (SCZ), bipolar disorder (BD), major depressive disorder (MDD), obsessive-compulsive disorder (OCD), and anorexia nervosa (AN). The MR assumptions are listed as follows: (1) the single nucleotide polymorphisms (SNPs) from genome-wide association studies (GWAS) applied as instrumental variables (IVs) are related to exposures; (2) IVs are not associated with the confounders; (3) IVs affect the risk of outcome only by the exposure [22].

### 2.2. Data Extraction

The available summarized data were gained from the open database (IEU OPEN GWAS PROJECT: https://gwas.mrcieu.ac.uk/ (accessed on 1 May 2022)). In order to avoid the bias of population heterogeneity, only the European population summarized data were adopted. Summary data on psychiatric disorders were extracted from the Psychiatric Genomics Consortium (PGC: https://www.med.unc.edu/pgc/ (accessed on 1 May 2022)). The detailed information of the GWAS datasets is described in Table 1.

#### 2.2.1. Genetic Associations with Oxidative Stress Injury Biomarkers

Genetic predictors for 11 oxidative stress injury biomarkers were obtained from the most up-to-date GWAS, namely, GST, CAT, SOD, GPX, UA, zinc, tocopherol, ascorbate, retinol, albumin, and bilirubin. Data of GST, CAT, SOD, and GPX were referred from the INTERVAL study [23]; tocopherol and albumin data were quoted from the Twins UK cohort and KORA study [24]; zinc data were based on a dataset generated by many different consortia developed for MR-Base [25]; the remaining data were gained from the UK biobank. The respective sample sizes were as follows: GST, 3301 individuals; CAT, 3301 individuals; SOD, 3301 individuals; GPX, 3301 individuals; UA, 343,836 individuals; zinc, 2630 individuals; tocopherol, 6266 individuals; ascorbate, 64,979 individuals; retinol, 62,911 individuals; albumin, 115,060 individuals; bilirubin, 342,829 individuals.

#### 2.2.2. Genetic Associations with Psychiatric Disorders

The summary GWAS data of 7 psychiatric disorders were obtained from the PGC website. As mentioned, only the European population summarized data were adopted to avoid population heterogeneity. The respective sample sizes were as follows: ASD (18,381 cases and 27,969 controls), ADHD (20,183 cases and 35,191 controls), SCZ (33,640 cases and 43,456 controls), BD (20,352 cases and 31,358 controls), MDD (170,756 cases and 329,443 controls), OCD (7037 cases and 33,925 controls), and AN (18,382 cases and 27,969 controls) (Table 1).

#### 2.2.3. Selection of IVs

Independent SNPs (r^2^ < 0.01 and distance > 250 kb) related to exposures were gained. The application of at least 10 independent SNPs as IVs can maintain sufficient statistical efficiency in the MR analysis [33], so the GWAS *p* value threshold was relaxed to 1 × 10^−5^, which allows for a sufficient number of SNPs. A larger *F* statistic indicated the stronger instrument strength, *F* statistics were used to test for weak instrumental variables. The *F* statistics of all SNPs included in the MR analysis were evaluated by mRnd (an online tool named, https://shiny.cnsgenomics.com/mRnd/ (accessed on 15 May 2022)), and all the *F* statistics of the included SNPs were more than 10.

### 2.3. Statistical Analysis

As the dominant analysis method, the random-effects inverse-variance weighted (IVW) method was applied to evaluate the causal association between the oxidative stress injury biomarkers and psychiatric disorders, and several sensitivity analyses were performed. Firstly, the weighted-median method was adopted to validate the associations. Then the MR pleiotropy residual sum and outlier (MR-PRESSO) test were introduced to explore the possible outliers and the corrected results were later calculated by removing outlier SNPs.

The odds ratios (ORs) and 95% confidence intervals (CIs) were used to present the causal effects of oxidative stress injury biomarkers on the risk of psychiatric disorders. Moreover, the causal effects of psychiatric disorders on the oxidative stress injury biomarkers were presented as beta and 95% CIs. All analyses were performed by R software (version 4.1.3) with *TwoSampleMR* and *MR-PRESSO* packages. A *p* value less than 0.0006 (0.05/77) was considered as statistically significant evidence of a causal association. A *p* value of less than 0.05 was considered as suggestive evidence for a potential causal association.

## 3. Results

### 3.1. Causal Effect of Genetically Predicted Oxidative Stress Injury Biomarkers on Psychiatric Disorders

Based on the Bonferroni-corrected threshold, there was no significant causal effect of oxidative stress injury biomarkers on psychiatric disorders (all *p* values were more than 0.0006). However, the results showed that part of the oxidative stress injury biomarkers was nominally associated with psychiatric disorders. Genetically predicted UA was associated with lower odds BD (OR = 0.999, 95% CI 0.997–1.000; *p* = 0.025), and higher odds ADHD (OR = 1.002, 95% CI 1.001–1.003; *p* = 0.004); CAT was associated with lower odds AN (OR = 0.903, 95% CI 0.816–0.999; *p* = 0.048); albumin was associated with higher odds ASD (OR = 1.162, 95% CI 1.035–1.304; *p* = 0.011) (Figure 2). By using the MR-PRESSO test, four outlier SNPs of BD, and two outliers of ADHD were detected. After correcting for possible outliers, the significance and magnitude of all these associations persisted were consistent (OR = 0.998, 95% CI 0.997–0.999; *p* = 8.10 × 10^−4^ for UA on BD; OR = 1.002, 95% CI 1.000–1.003; *p* = 0.006 for UA on MDD). Except for the CAT on AN (*p* of egger intercept = 0.008), none of the MR-Egger intercepts significantly deviated from 0, with *p* values of 0.178 for albumin on ASD, 0.791 for UA on BD, and 0.425 for UA on ADHD. The random effects model was used to evaluate the MR effect; the significance remained consistent. However, the relationship between UA and BD disappeared in the weighted-median method, and so did that between UA and ADHD. The detailed results could be viewed in the Appendix A.

### 3.2. Causal Effect of Genetically Predicted Psychiatric Disorders on Oxidative Stress Injury Biomarkers

Reverse MR analysis was conducted to investigate the potential causal effects of psychiatric disorders on oxidative stress injury biomarkers. A *p* value less than 0.0006 was considered as statistically significant evidence of a causal association, based on the Bonferroni-corrected threshold. A *p* value less than 0.05 was considered as suggestive evidence (Figure 3).

Results of the reverse MR analysis showed that there was a significant causal effect of MDD on total bilirubin after excluding one outliner SNP. The causal effect estimate was −0.144 (95% CI ranges from −0.184 to −0.105, *p* = 2.67 × 10^−4^), which is consistent in the weighted-median method (beta = −0.154, 95% CI ranges from −0.204 to −0.105, *p* = 0.002). The Egger intercept did not identify any pleiotropic SNP (*p* of egger intercept = 0.910). Moreover, genetically predicted ADHD showed a causal effect on ascorbate (beta = −0.034, 95% CI ranges from −0.054 to −0.033, *p* = 4.37 × 10^−5^), and there was no directional pleiotropy (*p* of egger intercept = 0.053). After we excluded 1 outliner SNP, the causal effect was still significant (beta = −0.039, 95% CI ranges from −0.049 to −0.029, *p* = 1.05 × 10^−4^), and the findings of the sensitivity analysis were consistent (Figure 3 and Figure 4).

There were nominal associations between BD and retinol (beta = 0.018, 95% CI ranges from 0.011 to 0.025, *p* = 0.014; *p* of egger intercept = 0.265), while the relationship disappeared in the weighted-median method. After excluding 13 outliner SNPs, BD was nominally associated with UA (beta = −0.656, 95% CI ranges from −0.983 to −0.328, *p* = 0.045; *p* of egger intercept = 0.503) and the relationship was consistent in the weighted-median method.

Suggestive evidence was also observed between MDD and UA with the beta of 1.793 (95% CI 1.078–2.509; *p* = 0.012) after excluding 11 outliner SNPs. The MDD was nominally associated with increased tocopherol (beta = 1.037, 95% CI ranges from 0.019 to 0.054, *p* = 0.034; *p* of egger intercept = 0.105) and decreased ascorbate (beta = −0.049, 95% CI ranges from −0.068 to −0.030, *p* = 0.009; *p* of the egger intercept = 0.461; the effect was consistent after excluding 1 outliner SNP, beta = −0.044, 95% CI ranges from −0.063 to −0.026, *p* = 0.018).

For SCZ, the IVW method yielded a nominal association of SCZ on decreased UA (beta = −0.964, 95% CI ranges from −1.261 to −0.667, *p* = 0.001; *p* of egger intercept = 0.001), the random effects model showed that this nominal association still existed; the effect was consistent after excluding 1 outliner SNP, beta = −0.793, 95% CI ranged from −1.305 to −0.552, *p* = 0.001) and increased retinol (beta = 0.017, 95% CI ranges from 0.011 to 0.023, *p* = 0.006; *p* of egger intercept = 0.151). Suggestive evidence was observed between SCZ and decreased albumin with the beta of 0.017 (95% CI ranges from 0.011 to 0.023, *p* = 0.005; *p* of egger intercept = 0.759).

As for ADHD, ADHD was nominally associated with decreased albumin (beta = −0.019, 95% CI ranges from −0.027 to −0.011, *p* = 0.015; *p* of egger intercept = 0.320), decreased bilirubin (beta = −0.042, 95% CI ranges from −0.011 to −0.063, *p* = 0.046; *p* of egger intercept = 0.685; this causal effect disappeared in weighted median method), and decreased CAT (beta = −0.098, 95% CI ranges from −0.142 to −0.055, *p* = 0.024; *p* of egger intercept = 0.101; this causal effect disappeared in weighted median method). There were also nominal causal effects on increased UA (four outliner SNPs, beta = 0.815, 95% CI ranges from 0.478 to 1.152; *p* of egger intercept = 0.772; the effect was consistent in the weighted median method) after excluding the outliner SNPs.

The AN was nominally associated with decreased UA (beta = −0.675, 95% CI ranges from −0.974 to −0.376, *p* = 0.024; *p* of egger intercept = 0.037, the random effects model showed that this nominal association still existed), while this relationship disappeared in the weighted median method. Other results could be viewed in the Appendix A.

## 4. Discussion

In the present study, a bidirectional MR analysis was performed to evaluate the causal associations between 11 oxidative stress injury biomarkers and 7 psychiatric disorders. Findings indicated that genetically predicted MDD was significantly associated with decreased total bilirubin and ADHD was significantly associated with decreased ascorbate. There was also suggestive evidence for a possible causal effect of UA on BD and ADHD, as well as CAT on AN and albumin on ASD. The nominally causal effects of psychiatric disorders on oxidative stress injury biomarkers were also detected. BD was nominally associated with decreased UA and increased retinol. MDD was nominally associated with increased UA and decreased ascorbate. SCZ was nominally associated with decreased UA and increased retinol and albumin. ADHD was nominally associated with increased UA, and decreased CAT, albumin, and bilirubin; AN was nominally associated with decreased UA.

Although numerous observational studies have emphasized the association between oxidative stress and psychiatric disorders, there were only a few related MR studies about it. These MR studies usually concentrated on UA and only performed unidirectional MR studies to detect the causal effects of UA on psychiatric disorders and turned out negative results [34,35]. In this work, the causal effects of oxidative stress injury biomarkers on psychiatric disorders were not significant. However, suggestive causal associations were found between UA and the risk of BD and ADHD, between CAT and the risk of AN, as well as between albumin and the risk of ASD. These associations point to the imbalance of peroxides and antioxidant defense as a potential cause of psychiatric disorders. Moreover, the antioxidative effects of antipsychotics, antidepressants, and mood stabilizers and the symptom-improving effects of supplements with antioxidants also emphasized the findings, i.e., the oxidative stress system plays a role in the pathogenesis of psychiatric disorders [36,37]. Larger epidemiological studies and directed laboratory studies are needed to determine the biochemical and chemical biological bases for these associations.

Although mitochondria can reduce the excessive reactive oxygen species, this ability is quite limited. Hence, antioxidants are essential for our body to counteract oxidative damage. The enzymatic antioxidants, such as GST, CAT, SOD, and GPX, could inhibit the formation of peroxide and remove the free radicals [38]. CAT, one of the enzymatic antioxidants, has the ability to reduce the H_2_O_2_ (SOD could catalyze the superoxide anion radical [O_2_] into H_2_O_2_) to the water and molecular oxygen, which protect the body against deleterious oxidation. In response to the increased production of radical oxygen species (ROS) and the associated oxidative damage, as a compensatory mechanism, the concentrations of antioxidative enzymes may increase. ADHD was found to be suggestively related to low levels of CAT, which is consistent with previous studies [39]. Furthermore, the methylphenidate for ADHD could alleviate the symptoms by increasing the activity of CAT [40].

Non-enzymatic antioxidants, including UA, albumin, bilirubin, zinc, tocopherol, ascorbate, and retinol, also play critical roles in the antioxidant system. They have the ability to chelate transition metals and interact with ROS by breaking free radical chain reactions [41]. UA, the end product of purine metabolism, is one of the peripheral non-enzymatic antioxidants and is related to the purinergic transformation. It can act on neurons, presynaptically and postsynaptically, which affect some neurotransmitter activities involved in psychosis, including dopamine, gamma-aminobutyric acid, glutamate, and serotonin [42]. UA was also reported to be associated with some specific traits, including driving and disinhibition, which are very common in BD and ADHD [43]. In this study, UA was associated with most psychiatric disorders. In our previous study, the abnormality of peripheral non-enzymatic antioxidants in schizophrenia was found, and results showed that patients with SCZ had a higher level of UA than healthy controls, while the difference in UA between antipsychotic-naïve patients and healthy controls was not significant. After receiving the antipsychotic treatment, the UA level in antipsychotic-naïve patients increased, which indicated that both the status of SCZ and antipsychotics could affect the oxidative stress system [44]. In this study, SCZ was nominally associated with low UA, which provides evidence for our previous study. An appropriately increased UA could enhance the antioxidant capacity to avoid oxidative damage, which might explain the results in this study that high-level UA was suggestively related to the low ratio BD. Moreover, UA was applied as a biomarker of BD, and several studies also suggested that UA could be used to distinguish BD and MDD, to which conclusion our previous study also reached [45]. In this study, MDD was associated with increased UA. On the contrary, BD was associated with decreased UA. This finding provided evidence that the UA was a potential biomarker for distinguishing between BD and MDD. Albumin is also a marker that contributes to inflammation in the immune system, which has been proven in the pathogenesis of psychiatric disorders. In this study, it was related to SCZ, ASD, and ADHD. As for TBIL, in addition to its antioxidant properties, it has a toxic effect on the brain, and evidence supports that it was related to cognition; the abnormal level of TBIL was also found in psychiatric disorders [46,47,48]. The result showed that MDD and ADHD were associated with decreased TBIL.

Retinol is vital to embryonic development, especially to the development of the brain. A large body of evidence indicates that retinol dysfunction (of the retinol process) is involved in the pathogenesis of psychiatric disorders, such as SCZ and ADHD [49]. In this study, retinol was associated with SCZ, ADHD, and BD. Ascorbate, as a neuroprotective compound, is highly concentrated in the brain and regulates the functioning of neurons and synapses [50]. In this study, ADHD was significantly associated with decreased ascorbate, and MDD was also related to decreased ascorbate.

There are several strengths in the present study. Firstly, 7 psychiatric disorders and 11 oxidative stress injury biomarkers were included, which made it the most comprehensive MR study between psychiatric disorders and the oxidative stress system. Secondly, the SNPs set as IVs were all obtained from a European population, minimizing the possibility of population stratification bias, and increasing the plausibility of the two-sample MR assumption. There were some limitations as well. Firstly, in order to include more SNPs to maintain the study power in MR, the threshold of the *p* value was set as 1 × 10^−5^, meaning the proportion of variance explained for the associations between IVs and some exposures was relatively small. Data derived from larger sample sizes would provide a more accurate assessment of the genetic influences on exposures, although the *F*-statistics showed that there were no weak IVs in this study. Secondly, the potential causal associations we reported on in the study should be interpreted with caution given that the *p* values were almost at nominal levels (only 2 *p* values were at significant levels), and the lack of associations should be interpreted carefully because of the limitations of MR. Finally, in this study, cumulative effects of exposure were throughout an individual’s lifetime, sadly we failed to uncover the possible effects of changes in oxidative stress status on the risk of psychiatric disorders.

## 5. Conclusions

In conclusion, this bidirectional MR study indicates that genetically predicted MDD was significantly associated with decreased total bilirubin, and ADHD was significantly associated with decreased ascorbate. It also provides evidence for the previous study, such as the abnormal levels of antioxidants in different psychiatric disorders, UA as a biomarker for precise diagnosis of BD and MDD, and the potential pathogenesis in psychosis. Moreover, it brings additional insight into the relationships between oxidative stress and psychiatric disorders, and the mechanism linking the oxidative stress system and psychiatric disorders is expected to be explored.

## Figures and Tables

**Figure 1 antioxidants-11-01386-f001:**
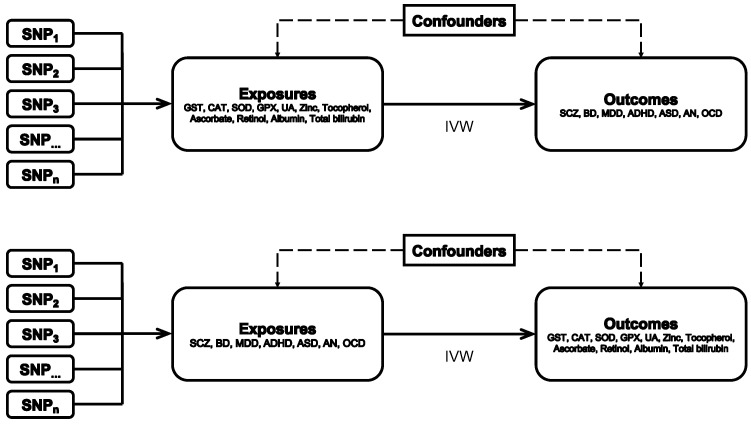
Flow chart of the bidirectional MR study design. GST, glutathione S-transferase; CAT, catalase; SOD, superoxide dismutase; GPX, glutathione peroxidase; UA, uric acid; SCZ, schizophrenia; BD, bipolar disorder; MDD, major depressive disorder; ADHD, attention-deficit/hyperactivity disorder; ASD, autism spectrum disorder; OCD, obsessive-compulsive disorder; AN, anorexia nervosa; IVW, inverse variance weighted method; SNP, single nucleotide polymorphism.

**Figure 2 antioxidants-11-01386-f002:**
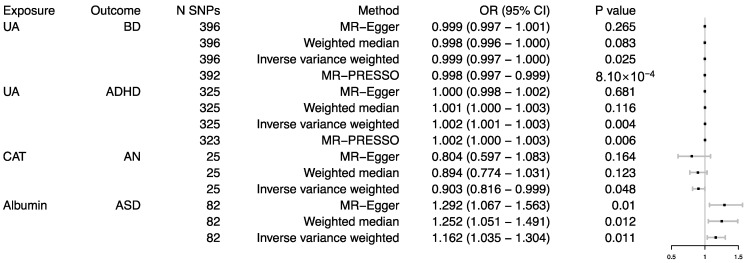
Associations between genetically predicted oxidative stress injury biomarkers and the risk of psychiatric disorders. CAT, catalase; UA, uric acid; BD, bipolar disorder; ADHD, attention-deficit/hyperactivity disorder; ASD, autism spectrum disorder; AN, anorexia nervosa; SNP, single nucleotide polymorphism.

**Figure 3 antioxidants-11-01386-f003:**
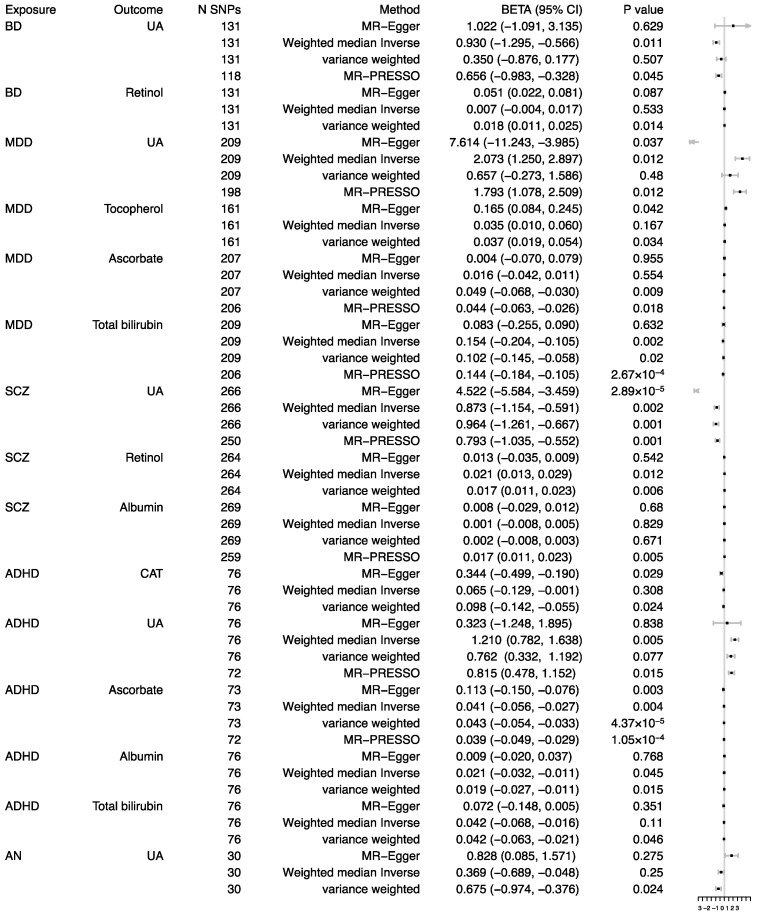
Associations between genetically predicted psychiatric disorders and oxidative stress injury biomarkers. CAT, catalase; UA, uric acid; SCZ, schizophrenia; BD, bipolar disorder; MDD, major depressive disorder; ADHD, attention-deficit/hyperactivity disorder; AN, anorexia nervosa; SNP, single nucleotide polymorphism.

**Figure 4 antioxidants-11-01386-f004:**
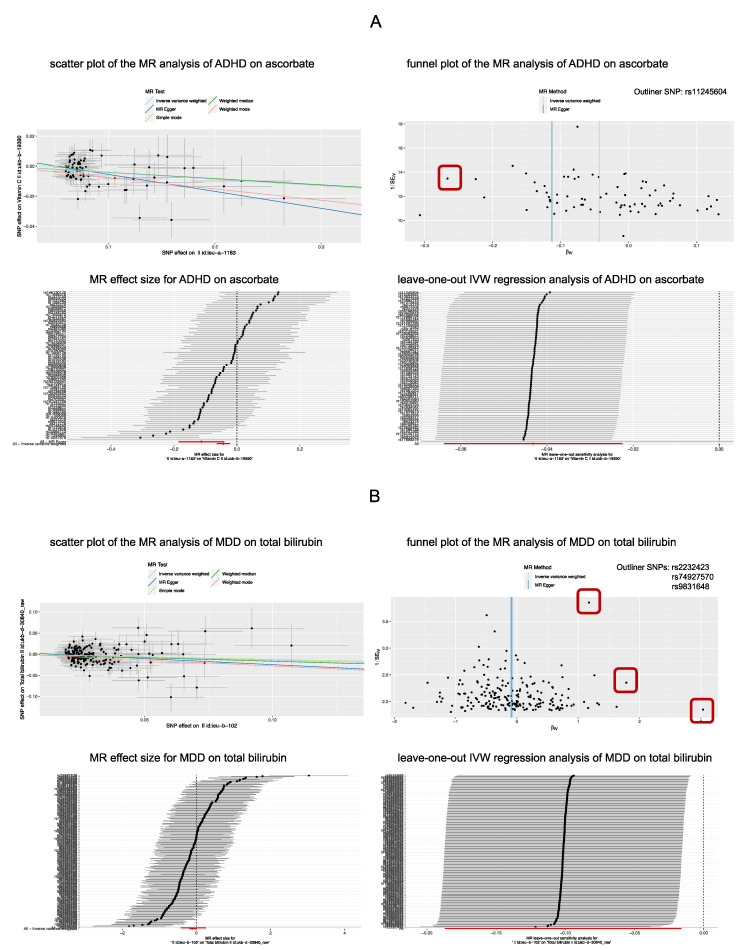
The significant causal effect of MDD on total bilirubin (**A**) and ADHD on ascorbate (**B**). (**A**) Significant causal effect of ADHD on ascorbate; (**B**) significant causal effect of MDD on total bilirubin. UA, uric acid; MDD, major depressive disorder; ADHD, attention-deficit/hyperactivity disorder.

**Table 1 antioxidants-11-01386-t001:** Detailed information regarding studies and datasets used in the present study.

Exposure or Outcome	Ref.	Ancestry	Participants	Web Source (accessed on 1 May 2022)
**Oxidative stress injury biomarkers**				
GST	[23]	European	3301 individuals	https://gwas.mrcieu.ac.uk/datasets/prot-a-1283/
CAT	[23]	European	3301 individuals	https://gwas.mrcieu.ac.uk/datasets/prot-a-367/
SOD	[23]	European	3301 individuals	https://gwas.mrcieu.ac.uk/datasets/prot-a-2800/
GPX	[23]	European	3301 individuals	https://gwas.mrcieu.ac.uk/datasets/prot-a-1265/
UA	/	European	343,836 individuals	https://gwas.mrcieu.ac.uk/datasets/ukb-d-30880_raw/
Tocopherol	[24]	European	6266 individuals	https://gwas.mrcieu.ac.uk/datasets/met-a-571/
Zinc	[25]	European	2630 individuals	https://gwas.mrcieu.ac.uk/datasets/ieu-a-1079/
Ascorbate	/	European	64,979 individuals	https://gwas.mrcieu.ac.uk/datasets/ukb-b-19390/
Retinol	/	European	62,911 individuals	https://gwas.mrcieu.ac.uk/datasets/ukb-b-17406/
Albumin	/	European	115,060 individuals	https://gwas.mrcieu.ac.uk/datasets/met-d-Albumin/
Total bilirubin	/	European	342,829 individuals	https://gwas.mrcieu.ac.uk/datasets/ukb-d-30840_raw/
**Psychiatric disorders**				
Schizophrenia	[26]	European	33,640 cases and 43,465 controls	https://www.med.unc.edu/pgc/download-results/
Bipolar disorder	[27]	European	20,352 cases and 31,358 controls	https://www.med.unc.edu/pgc/download-results/
Major depressive disorder	[28]	European	170,756 cases and 329,443 controls	https://www.med.unc.edu/pgc/download-results/
Autism spectrum disorder	[29]	European	18,381 cases and 27,969 controls	https://www.med.unc.edu/pgc/download-results/
Attention-deficit/hyperactivity disorder	[30]	European	20,183 cases and 35,191 controls	https://www.med.unc.edu/pgc/download-results/
Obsessive-compulsive disorder	[31]	European	7037 cases and 33,925 controls	https://www.med.unc.edu/pgc/download-results/
Anorexia nervosa	[32]	European	18,382 cases and 27,969 controls	https://www.med.unc.edu/pgc/download-results/

Note: GST, glutathione S-transferase; CAT, catalase; SOD, superoxide dismutase; GPX, glutathione peroxidase; UA, uric acid; Ref.: References.

## Data Availability

This study was based on publicly available summarized data (IEU OPEN GWAS PROJECT: https://gwas.mrcieu.ac.uk/ (accessed on 1 May 2022)) and PGC (https://www.med.unc.edu/pgc/ (accessed on 1 May 2022)).

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
