# Peer review of "Oxidative Stress and Psychiatric Disorders: Evidence from the Bidirectional Mendelian Randomization Study"

_antioxidants, 2022, doi:10.3390/antiox11071386_

Round 1

Reviewer 1 Report

Thank you for this contribution. The links connecting oxidative stress and neurodegenerative disorders are compelling by now, and there is suggestive evidence oxidative stress and psychiatric disorder also are intertwined. But it is not certain if the contributions of oxidative stress are independent of genetic contributions, or act along the same biological pathways. Because the contributions of most of the genetic and non-genetic risk factors to psychiatric disorders have small effect sizes, this is not an  easy question to address. This study is treating molecules that contribute to oxidative stress, by their presence or absence in the subject's environment, as modifiable risk factors.  However, the association between oxidative stress and psychiatric disorders might be confounded by possible influence of an unidentified precedent factor(s) that causes the changes in the oxidative stress markers you are using. 

There are two additional points I would like to see addressed: 

1. The connection of this study with other analysis of the potential roles of oxidative stress in the pathogenesis of psychiatric disorders;

2. Albert Einstein is famous for suggesting in his papers lab experiments that experimentalists should take. In a similar vein, I think it support the worthiness of your publication if you could point to new studies that should be undertaken as a result of your work.

Additionally, there are a few sentences that are not correct English. Namely, lines 37, 52, 55, 59, 61, 65, 72, 78, 85, 101, 119, 178, 200, 233, 261, and 264. In most cases, the sentences were incomplete or were not written in standard English, or possessed inaccurate phrasing. Everything else is well-written, so I'm sure you will spot the problem and fix them with no problems. 

Author Response

Manuscript ID: antioxidants-1771105

Title: Oxidative stress and psychiatric disorders: evidence from the bidirectional Mendelian randomization study

Response to Reviewer’s Comments

Thanks for your constructive comments and suggestions regarding our manuscript. The following is our response to the review. For clarity, we have directly quoted or paraphrased the reviewers’ specific concerns or critiques in italics, then provided our response and described related revisions. All revisions are highlighted in the revised manuscript using yellow color.

Comment 1

The connection of this study with other analysis of the potential roles of oxidative stress in the pathogenesis of psychiatric disorders;

Response 1

Thanks for your constructive comments, and we added the related discussion about the relationship between other analyses of the of the potential roles of oxidative stress in the pathogenesis of psychiatric disorders in Discussion section (Line 271-324).

Comment 2

Albert Einstein is famous for suggesting in his papers lab experiments that experimentalists should take. In a similar vein, I think it support the worthiness of your publication if you could point to new studies that should be undertaken as a result of your work.

Response 2

Thanks for your kindly reminder, idea of this new study did originate from our previous study. In our previous studies, abnormal levels of antioxidants were found in schizophrenia, bipolar disorders and major depressive disorder. Hence, we conducted this bidirectional Mendelian randomization study to validate our previous conclusions, and the results were confirmed by this study. We also discussed it in the Discussion section (Line 296-317).

Comment 3

Additionally, there are a few sentences that are not correct English. Namely, lines 37, 52, 55, 59, 61, 65, 72, 78, 85, 101, 119, 178, 200, 233, 261, and 264. In most cases, the sentences were incomplete or were not written in standard English, or possessed inaccurate phrasing. Everything else is well-written, so I'm sure you will spot the problem and fix them with no problems.

Response 3

Thank you very much for pointing out our language mistakes. All the mistakes you mentioned have been revised as suggested. We have highlighted all changes in the revised manuscript using yellow color.

Reviewer 2 Report

The purpose of the article is clear, and the method is itself adequate to provide an overview of the issue. Nevertheless, the overall perspective is that of saying that a general model of a disease (oxidative stress) indicates some disease, with variable correlation within the same area of biological abnormality. Thereby, authors run the risk of clarifying in methodologically sound way a generic finding about the concept of disease. We suggest that authors discuss deeper whether the single findngs about each diagnosis may lead to a different interpretation of some specific pathway, or not. Also, one may object that if one parameter turns out to be related to a diagnosis out of a list of 11 putative parameters, the correlation with oxidative stress is feeble, if any. It would be then more meaningful to focus on specific findings rather than discussing on a general positive correlation with oxidative stress: positive for at least one parameter for some diagnoses, but not so generalized at all for all diagnoses and all parameters.

Lastly, I would just avoid to mention findings which are not statistically significant, otherwise the main point of strenght of the study method is somehow neutralized by suggestions which cannot count on statistical support.

The conclusions should be amended consistently with the above corrections.

Author Response

Manuscript ID: antioxidants-1771105

Title: Oxidative stress and psychiatric disorders: evidence from the bidirectional Mendelian randomization study

Response to Reviewer’s Comments

Thanks for your constructive comments and suggestions regarding the above-referenced manuscript. The following is our response to the review. For clarity, we have directly quoted or paraphrased the reviewers’ specific concerns or critiques in italics, then provided our response and described related revisions. All revisions are highlighted in the revised manuscript using yellow color.

Comment 1

The purpose of the article is clear, and the method is itself adequate to provide an overview of the issue. Nevertheless, the overall perspective is that of saying that a general model of a disease (oxidative stress) indicates some disease, with variable correlation within the same area of biological abnormality. Thereby, authors run the risk of clarifying in methodologically sound way a generic finding about the concept of disease. We suggest that authors discuss deeper whether the single findings about each diagnosis may lead to a different interpretation of some specific pathway, or not. Also, one may object that if one parameter turns out to be related to a diagnosis out of a list of 11 putative parameters, the correlation with oxidative stress is feeble, if any. It would be then more meaningful to focus on specific findings rather than discussing on a general positive correlation with oxidative stress: positive for at least one parameter for some diagnoses, but not so generalized at all for all diagnoses and all parameters.

Response 1

Thanks for your constructive comments and suggestions, which help us display and explain our results deeper and more clearly. We discuss deeper about each finding and added the related contents about the specific mechanism and pathway of each oxidative markers. (Line 273-326)

Comment 2

Lastly, I would just avoid to mention findings which are not statistically significant, otherwise the main point of strength of the study method is somehow neutralized by suggestions which cannot count on statistical support.

The conclusions should be amended consistently with the above corrections.

Response 2

As your suggestions, we reorganized our conclusion and make it consistently with the above corrections. (Line 345-350)

Reviewer 3 Report

Review of a manuscript “Oxidative stress and psychiatric disorders: evidence from the bidirectional Mendelian randomization study” by Zhe Lu and coauthors submitted to “Antioxidants”.

Oxidative stress plays an essential and not completely understood role in many human diseases, including psychiatric disorders. In many cases it is difficult to understand if oxidative stress injury is a cause, or a downstream effect of human diseases. The authors used Mendelian randomization and the genetic variants from genome-wide association studies in order to clarify the casual relationship and better understand the role of oxidative stress in psychiatric disorders.  This is an interesting direction of biomedical studies and the results presented in the manuscript will be interesting for the readership of “Antioxidants”.

 The following corrections and additions should be made in the manuscript:

 Introduction:

Line 37: “Psychiatric disorders are public health issues of the world-wide people” this is an awkward sentence, which should be corrected, for example, as  “Psychiatric disorders are public health issue all over the world”

Line 43 -47: “Dysfunction of neurotransmission (for instance, dopaminergic, serotoninergic and glutamatergic neurotransmission) appears to contribute to the genesis of psychiatric disorders, but the evidence also points to a more widespread and variable involvement of other central nervous system and peripheral system, such as neurotrophic factors, immune system…”

Hard to read sentence; bad style of presentation can be corrected as follows: ”Dysfunction of dopaminergic, serotoninergic and glutamatergic neurotransmission contributes to the genesis of psychiatric disorders, however, more often other components, such as neurotrophic factors, immune system…”

Line 50: “The balanced oxidative stress status is essential for the normal function of the body.” After this sentence the authors should add the following sentence and citation: ”High level of oxidation might lead to oxidative changes of proteins playing an important role in human diseases” Reference: “gamma-Synuclein: Seeding of α-Synuclein Aggregation and Transmission  Between Cells. Biochemistry, 2012; 51(23):4743-54”

 Lines 65-66: ”Compared with health individuals, patients with SCZ were associated with lower GSH, lower total antioxidant status…” should be corrected as follows: ”Compared with healthy individuals, patients with SCZ have lower GSH, lower total antioxidant status…”

Materials and methods.

Lines 91-92: ”The oxidative stress injury biomarkers was composed of glutathione S-transferase (GST), catalase (CAT)..should be corrected as ”A group of oxidative stress injury biomarkers was composed of glutathione S-transferase (GST), catalase (CAT)…”

 Results

3.1. “Genetically predicted oxidative stress injury biomarkers on psychiatric disorder” this is not completed sentence, the sense of which is unclear. Something is missing here :Effect? Role?

Discussion

Lines 264-267: Too long and hard to read sentence with mistakes in style and grammar: “In our study, the causal effects of oxidative stress injury biomarkers on psychiatric disorders did not be significant, but there were suggestive causal association between UA and the risk of BD and ADHD, between CAT of the risk of AN, as well as between albumin and the risk of ASD, which indicated that the imbalance of peroxides and antioxidant defense might be a potential cause of psychiatric disorders.” Should be rewritten as follows:” In our study, the causal effects of oxidative stress injury biomarkers on psychiatric disorders were not significant. However, we found suggestive causal associations between UA and the risk of BD and ADHD, between CAT of the risk of AN, as well as between albumin and the risk of ASD. These associations point to the imbalance of peroxides and antioxidant defense as a potential cause of psychiatric disorders”

Author Response

Manuscript ID: antioxidants-1771105

Title: Oxidative stress and psychiatric disorders: evidence from the bidirectional Mendelian randomization study

Response to Reviewer’s Comments

Thanks for your constructive comments and suggestions regarding the above-referenced manuscript. The following is our response to the review. For clarity, we have directly quoted or paraphrased the reviewers’ specific concerns or critiques in italics, then provided our response and described related revisions. All revisions are highlighted in the revised manuscript using yellow color.

Comment 1

Line 37: “Psychiatric disorders are public health issues of the world-wide people” this is an awkward sentence, which should be corrected, for example, as “Psychiatric disorders are public health issue all over the world”

Response 1

Thank you very much for pointing our grammatical error, and we have corrected it as your suggestion. (Line 37)

Comment 2

Line 43 -47: “Dysfunction of neurotransmission (for instance, dopaminergic, serotoninergic and glutamatergic neurotransmission) appears to contribute to the genesis of psychiatric disorders, but the evidence also points to a more widespread and variable involvement of other central nervous system and peripheral system, such as neurotrophic factors, immune system…”

Hard to read sentence; bad style of presentation can be corrected as follows: “Dysfunction of dopaminergic, serotoninergic and glutamatergic neurotransmission contributes to the genesis of psychiatric disorders, however, more often other components, such as neurotrophic factors, immune system…”

Response 2

Thanks for your comment, and we have corrected it as your suggestion. (Line 43-37)

Comment 3

Line 50: “The balanced oxidative stress status is essential for the normal function of the body.” After this sentence the authors should add the following sentence and citation: “High level of oxidation might lead to oxidative changes of proteins playing an important role in human diseases” Reference: “gamma-Synuclein: Seeding of α-Synuclein Aggregation and Transmission Between Cells. Biochemistry, 2012; 51(23):4743-54”

Response 3

Thanks for your comment, and we have added the sentence and reference as your suggestion. (Line 48-50, reference 3)

Comment 4

Lines 65-66: “Compared with health individuals, patients with SCZ were associated with lower GSH, lower total antioxidant status…” should be corrected as follows: “Compared with healthy individuals, patients with SCZ have lower GSH, lower total antioxidant status…”

Response 4

Thanks for your comment, and we have corrected it as your suggestion. (Line 65-66)

Comment 5

Lines 91-92: “The oxidative stress injury biomarkers was composed of glutathione S-transferase (GST), catalase (CAT)..should be corrected as ”A group of oxidative stress injury biomarkers was composed of glutathione S-transferase (GST), catalase (CAT)…”

Response 5

Thanks for your comment, and we have corrected it as your suggestion. (Line 90-92)

Comment 6

3.1. “Genetically predicted oxidative stress injury biomarkers on psychiatric disorder” this is not completed sentence, the sense of which is unclear. Something is missing here: Effect? Role?

Response 6

Thanks for your comment, and we have corrected the subtitle 3.1 and 3.2 as your suggestion.

Comment 7

Lines 264-267: Too long and hard to read sentence with mistakes in style and grammar: “In our study, the causal effects of oxidative stress injury biomarkers on psychiatric disorders did not be significant, but there were suggestive causal association between UA and the risk of BD and ADHD, between CAT of the risk of AN, as well as between albumin and the risk of ASD, which indicated that the imbalance of peroxides and antioxidant defense might be a potential cause of psychiatric disorders.” Should be rewritten as follows:” In our study, the causal effects of oxidative stress injury biomarkers on psychiatric disorders were not significant. However, we found suggestive causal associations between UA and the risk of BD and ADHD, between CAT of the risk of AN, as well as between albumin and the risk of ASD. These associations point to the imbalance of peroxides and antioxidant defense as a potential cause of psychiatric disorders”

Response 7

Thanks for your comment, and we have corrected it as your suggestion. (Line 265-269)

Round 2

Reviewer 1 Report

My enthusiasm for this manuscript is tempered for two reasons. First, your results contibute suggestive evidence that oxidative stress and psychiatric disorders are linked. However, there is no evidence the connection is causal or that that this association is unique for psychiatric disorders, but I sense you would like to draw a stronger conclusion. It could be that I am mis-reading your conclusions. This takes me to my second point. In my initial review, I indicated that moderate English changes were required, and it is clear you attempted to improve the English in this draft. However you are still falling short of what is acceptable, and I have therefore changed my recommendation to 'Extensive editing of English language...'. There are still too many run-on sentences, sentences with apparently inverted phrases, sentences with words missing, or with incorrect words. As an example of the latter, in presenting the CI ranges, you most often used the word 'arrange', as in "CI arrange from X to Y" (lines 189, 191, 193, 196, ...). If the manuscript were to be published in its current form it would not have the full impact that it otherwise might if these short-comings were fixed.

Author Response

Manuscript ID: antioxidants-1771105

Title: Oxidative stress and psychiatric disorders: evidence from the bidirectional Mendelian randomization study

We thank the reviewer's constructive comments and suggestions regarding the above-referenced manuscript. The following is our response to the review. For clarity, we have directly quoted or paraphrased the reviewers’ specific concerns or critiques in italics, then provided our response and described related revisions. All revisions are highlighted in the revised manuscript using yellow color.

Comment 1

First, your results contibute suggestive evidence that oxidative stress and psychiatric disorders are linked. However, there is no evidence the connection is causal or that that this association is unique for psychiatric disorders, but I sense you would like to draw a stronger conclusion. It could be that I am mis-reading your conclusions. 

Response 1

Sorry for confusing you. In this study, we firstly made an assumption that the oxidative stress might be a potential cause of psychiatric disorders, so we set the oxidative stress injury biomarkers as exposure, and set the psychiatric disorders as outcomes. If the results were significant, it could prove the above assumption. Unfortunately, no result was significant after adjusting (P < 0.0006), only some nominal results were found. Then, we want to explore the effect of psychiatric disorders on oxidative stress status, so we performed the reverse MR analysis. The results showed that MDD was significantly associated with decreased bilirubin (P=2.67E-04) and ADHD was significantly associated with decreased ascorbate (P=4.37E-05). Some nominal results were also found in the reverse MR study. According to the explanation of MR study, the significant result could be interpreted as the causal effect of the exposure on outcomes was significant. Therefore, we came to the conclusion.

Comment 2

 In my initial review, I indicated that moderate English changes were required, and it is clear you attempted to improve the English in this draft. However you are still falling short of what is acceptable, and I have therefore changed my recommendation to 'Extensive editing of English language...'. There are still too many run-on sentences, sentences with apparently inverted phrases, sentences with words missing, or with incorrect words. As an example of the latter, in presenting the CI ranges, you most often used the word 'arrange', as in "CI arrange from X to Y" (lines 189, 191, 193, 196, ...). If the manuscript were to be published in its current form it would not have the full impact that it otherwise might if these short-comings were fixed.

Response 2

We are so sorry for our poor language. After receiving your comments, we invited a colleague major in English interpreting to revised our manuscript. We have highlighted all changes in the revised manuscript using yellow color.

Reviewer 2 Report

ok, amended as suggested.

Author Response

Thanks for your review of our manuscript!

Reviewer 3 Report

The authors improved the manuscript by answering the reviewer's questions and comments

Author Response

(The authors gave the same response as above.)

Round 3

Reviewer 1 Report

There are still grammatical issues in the paper, but these are minor and can be easily resolved. First, in lines 154-157 are  confused. I think you meant to say:  "A P-value less than 0.0006 (0.05/77) was considered as statistically significant evidence of a causal association. A P-value less than 0.05 was considered as suggestive evidence for a potantial causal association." Second, in reporting the results Section 3.2, please adopt the same convention that you utilized in Section 3.1.

My main concern this time around is with your discussion. The second paragraph of the discussion is the most significant one, in which you assert that this study showed suggestive evidence that UA is causally associated with risk for BD and ADHA; CAT with AN; albumin and risk for ASD. I have no problems with this. I believe you should then point out that larger epidemiological studies and directed laboratory studies are needed to determine the biochemical and chemical biological basis for these associations.

However, you go on to speculate how oxidative damage, generally, and certain analytes/biomarkers, specifically, contributes to psychiatric damage. Oxidative damage affects processes at every developmental stage in every organizational level in the body, and it is certainly a common thread that runs through many diseases, including psychiatric and neurodegenerative diseases. Because of this far reach, and without reference to reliable, manipulable model systems, I don't think it is useful at this stage to speculate too deeply about the mechanistic implications of your findings.  Moreover, at this point you make statements that are simplistic almost to the point of being tautological (lines 280 - 282), or you make generalizations that are too sweeping (you assert UA plays a role in some physiological function...lines 293 - 296, when it may, or it may simply be a by-product), or you make assertions that you have not proven (for instance, your evidence does not 'prove' that increased UA enhances antioxidant capacity to avoid oxidative damage (lines 304 - 306)).  For these reasons, I urge you to use caution and restraint in your discussion.

Author Response

Manuscript ID: antioxidants-1771105

Title: Oxidative stress and psychiatric disorders: evidence from the bidirectional Mendelian randomization study

We thank the reviewer's constructive comments and suggestions regarding the above-referenced manuscript. The following is our response to the review. For clarity, we have directly quoted or paraphrased the reviewers’ specific concerns or critiques in italics, then provided our response and described related revisions. All revisions are highlighted in the revised manuscript using yellow color.

Comment 1

There are still grammatical issues in the paper, but these are minor and can be easily resolved. First, in lines 154-157 are confused. I think you meant to say: "A P-value less than 0.0006 (0.05/77) was considered as statistically significant evidence of a causal association. A P-value less than 0.05 was considered as suggestive evidence for a potantial causal association." Second, in reporting the results Section 3.2, please adopt the same convention that you utilized in Section 3.1.

Response 1

Thanks for your helpful comments. Sorry for our misleading expression and we have corrected the sentence as your suggestion. (lines 154-157).

We also added the contents about the P-value threshold at the beginning of the Section 3.2 (lines 189-192) and unified the expression of the results in Section 3.2.

Comment 2

My main concern this time around is with your discussion. The second paragraph of the discussion is the most significant one, in which you assert that this study showed suggestive evidence that UA is causally associated with risk for BD and ADHA; CAT with AN; albumin and risk for ASD. I have no problems with this. I believe you should then point out that larger epidemiological studies and directed laboratory studies are needed to determine the biochemical and chemical biological basis for these associations.

Response 2

Thank you for the constructive advice. We have added the related content as your helpful suggestion. (lines 274-276)

Comment 3

However, you go on to speculate how oxidative damage, generally, and certain analytes/biomarkers, specifically, contributes to psychiatric damage. Oxidative damage affects processes at every developmental stage in every organizational level in the body, and it is certainly a common thread that runs through many diseases, including psychiatric and neurodegenerative diseases. Because of this far reach, and without reference to reliable, manipulable model systems, I don't think it is useful at this stage to speculate too deeply about the mechanistic implications of your findings.  Moreover, at this point you make statements that are simplistic almost to the point of being tautological (lines 280 - 282), or you make generalizations that are too sweeping (you assert UA plays a role in some physiological function...lines 293 - 296, when it may, or it may simply be a by-product), or you make assertions that you have not proven (for instance, your evidence does not 'prove' that increased UA enhances antioxidant capacity to avoid oxidative damage (lines 304 - 306)).  For these reasons, I urge you to use caution and restraint in your discussion.

Response 3

Thanks for pointing our inappropriate discussion. Sorry for some unreasonable assertions. We carefully reviewed the Discussion section after receiving your comments and corrected the statements. We deleted the simplistic and tautological statements, and rewrote the sweeping generalizations (lines 296-297) to highlight the previous findings for UA in the psychiatric disorders. We also rewrote the erroneous expression (lines 306-307) to make it more accurate.

Round 4

Reviewer 1 Report

Thank you for addressing my concerns.